# REVISTING NEGATIVE TRANSFER USING ADVERSARIAL LEARNING

## ABSTRACT

An unintended consequence of feature sharing is the model fitting to correlated tasks within the dataset, termed negative transfer. In this paper, we revisit the problem of negative transfer in multitask setting and find that its corrosive effects are applicable to a wide range of linear and non-linear models, including neural networks. We first study the effects of negative transfer in a principled way and show that previously proposed counter-measures are insufficient, particularly for trainable features. We propose a adversarial training approach to mitigate the effects of negative transfer by viewing the problem in a domain adaptation setting. Finally, empirical results on attribute prediction multi-task on AWA and CUB datasets further validate the need for correcting negative sharing in an end-to-end manner.

## 1 INTRODUCTION

Advances in machine learning have led to proficient supervised learning models with powerful representations in various prediction tasks. We now expect an *ideal* classification model to restrict itself to a pertinent set of evidences available to it from the input for prediction. Further, we expect the model to disregard any unrelated evidences in the data to enable better generalization.

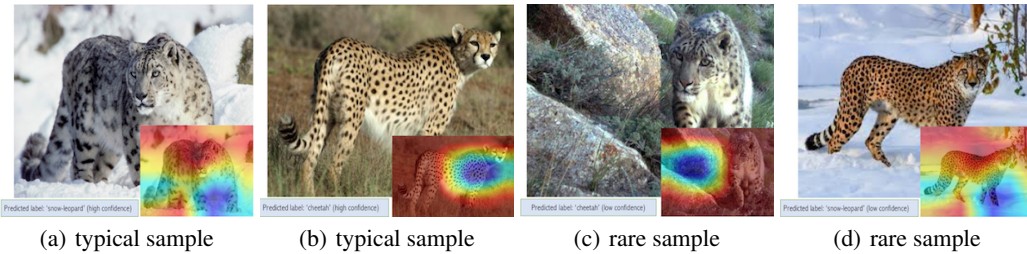

|  (a) typical sample  |  (b) typical sample  |  (c) rare sample  |  (d) rare sample  |

Figure 1: A supervised classifier 'cheetah *vs.* snow-leopard' that uses unrelated evidence (of habitat) over relevant evidence (of fur patterns). As shown by the pixel importance maps, the model suffers from the *negative transfer* prevalent in a typical animal image dataset skewed towards the animal's typical habitat and fails to generalize to rare samples.

Let us consider the task of training an animal classifier "cheetah *vs.* snow-leopards" from a dataset of images of these animals, such as those illustrated in Figure 1 – a task which ideally should focus on the animal's appearance features. However, a large portion of these images also contain various cues of the typical habitat of the animals in the background, i.e., tall grass and snow (see Figures 1 (a) and (b)) which are, in principle, unrelated to the animal's appearance. An archetypal model is deceived by the co-occurrence of such unrelated, yet easily detectable cues of habitat over the animal's appearance features such as complex fur patterns. However, a proficient supervised learning model must identify relevant evidences for the label of interest and at the same time discard various *unrelated* evidences such as presence of snow, even though it tends to co-occur frequently with snow-leopard. Consequently, it would be more likely that such a model would perform better on rare-instances (such as those in Figures 1 (c) and (d)) and generalize better to unseen instances.

This phenomenon of co-occurring but unrelated evidences being present in training data and thereby having a debilitating effect on model performance has been described in literature (Jayaraman et al. (2014); Zhou et al. (2013; 2010); Romera-Paredes et al. (2012); Yoon & Hwang (2017); Zhou et al. (2011)). These techniques utilize the easy availability of labels for unrelated evidences (e.g. background habitat labels above), called *negative* labels which constitutes an auxiliary task, and seek to mitigate its debilitating performance on the primary task (e.g. animal classification above) with techniques referred to as negative-sharing or negative-transfer.

While all of these techniques have tackled this problem utilizing various forms of regularization, we describe several shortcomings of this class of approaches, most notable of which is their inapplicability to the popular paradigm of trainable features obtained via neural representation learning. Motivated by these limitations, in this paper we depart from the direction of regularization-based approaches and examine methods inspired from a domain-adaptation viewpoint to propose an adversarial training-based formulation. We uniquely view such a scenario as an instance of adversarial multi-task learning, where the classification tasks are either the *primary* task of interest (i.e., predicting the presence of fur pattern and color) or the *auxiliary* negative tasks (i.e., characteristics of habitat) to be avoided. Since the 2 tasks are unrelated, any label correlation between primary and auxiliary labels in the training data is only by chance and therefore from a domain-adaptation perspective, we envision a target-domain as possibly having a *different* correlation between the primary and auxiliary labels. The effects of negative transfer are hence mitigated when the classification task is trained in this domain.

We discuss advantages of our proposed formulation, inspired from domain-adaptation, to alleviate the negative transfer over existing techniques, including ready applicability to neural networks in an end-to-end fashion. It must be noted that, while the formulation of the problem is motivated with multi-task learning, negative-transfer is a disposition of any supervised learning task from simple binary classification to recent popular supervised tasks such as image detection, captioning, or visual dialog. We present motivating literature that prelude this work next.

## 2 RELATED WORK ON TACKLING NEGATIVE SHARING

Image classification literature often characterize mid-level features as attribute predictors in multi-label learning. The inability of models to learn predictors with the correct semantic concept is attributed to negative transfer. To our knowledge, the predominent approach to tackle negative transfer in such setting was the use of specific regularizers (Romera-Paredes et al., 2012; Jayaraman et al., 2014). Specifically, the *primary* model avoid using features which are important for the *auxiliary* task, leading to models *competing* for features. Jayaraman et al. (2014) further extends this idea to attribute *groups*, where feature selection sparsity is induced across group, but encourage within them. We highlight three limitations of feature competition techniques below:

- Repeated features: Consider the simple scenario where some features in the feature representations are repeated or dependent on others. Feature competition enforces tasks to pick unique features, however they are implicitly the same. Going back to the case of 'cheetah vs snow leopard' example mentioned earlier, when there are two copies of a feature which captures 'snow in the background', then both primary and auxiliary classifiers would just pick different copies of that feature. Here, the idea of resisting cheetah/snow leopard classifier from picking features which capture snow is negated.

- Trainable features: Neural representations have become extremely popular owing the power of learning trainable features. However, feature competition techniques fail with prediction models with trainable features. If there are features that are important for both primary and auxiliary task, models involving a trainable feature setup will lead to duplicating the same features, thereby resulting in the *repeated feature* scenario.

- Easy auxiliary task with abundant features: Consider a scenario of an easy auxiliary task, that does not require a large number of features to be predicted correctly. Similar to the previous case, the spared features from the auxiliary task can be picked by the primary task resulting in negative transfer.

Motivated by these shortcomings, we proceed to examine the negative transfer problem in a domain adaptation setting.

## 3 PROPOSED FRAMEWORK

In this section we first formalize and explain the problem setting of negative transfer. We present scenarios where earlier attempts (regularization-based approaches) to tackle this problem fails. We then explain our formulation for posing then negative transfer problem in a domain adaptation setting and derive adversarial learning algorithms to solve negative transfer. Lastly, we present a thorough analysis of the proposed formulation to address negative transfer by experimenting on a carefully designed synthetic dataset.

### 3.1 PROBLEM SETUP

In typical 2-label classification problem, we assume that training data and all future test examples are drawn i.i.d from a probability distribution $D$ on instance space $\mathcal{X}$ and a labeling function $f_p :$ $\mathcal{X} \to \{+1, -1\}$ for the primary task labels and $f_a : \mathcal{X} \to \{+1, -1\}$ for the auxiliary task labels. Every instance $x \in \mathcal{X}$ has primary and auxiliary labels: $y_p$ and $y_a$ respectively. The goal is to learn a classifier which performs well on future samples from $D$, which may have a different label correlation since the tasks are unrelated. Formally, we capture this label correlation via a joint label distribution $P(Y) = P(Y_a, Y_b)$, and we assume that $P(Y)$ in training and test are different.

This problem setup is different from earlier works on negative transfer in the way how unrelated tasks are defined. We define unrelated tasks as the ones which can have different correlation in training and test data. In (Romera-Paredes et al., 2012) unrelated tasks is referred as tasks which are defined over orthogonal sets of features. Zhou et al. (2010) uses the term *negative correlation* for unrelatedness among tasks. Two tasks are negatively correlated when one feature is deemed to be *important* to first task makes it unlikely to be important to the second task.

### 3.2 NEGATIVE TRANSFER AS A DOMAIN ADAPTATION PROBLEM

Let the instances in training data be drawn from source distribution $D_S$. The aim is to train a classifier on this which performs well on instances drawn from target distribution $D_T$. Instances in $D_S$ has primary and auxiliary labels correlated with joint label distribution $P_S(Y) = P_S(Y_p, Y_a)$ whereas in $D_T$ the labels have a different correlation with different joint label distribution $P_T(Y) = P_T(Y_p, Y_a)$. We assume that the labelling function $f_p$ and $f_a$ are universal and common for both source and target domains.

Typical unsupervised domain adaptation setting has labelled instances from source distribution and *also* unlabelled instances from target distribution. However, we have no information (neither labels nor instances) about the target distribution. The only information we have about the target domain is $P_T(Y)$. We are either provided an estimate of $P_T(Y)$ from an oracle, or we make an assumption on $P_T(Y)$ (for instance, a uniform distribution over the space of $Y$).

Consider $\mathcal{U}_T$, the space of all distributions over $\mathcal{X}$ which has label distribution $P_T(Y)$. It is extremely challenging to adapt from the source domain to all members of $\mathcal{U}_T$ with given $P_T(Y)$. However, we can settle for a particular $D_T$. We pick such a $D_T \in \mathcal{U}_T$ as that distribution over $\mathcal{X}$ which is *nearest* to $D_S$. We model this in terms of *KL divergence* as a constrained optimization as follows.

$$D_T^\star = \arg\min_{D \in \mathcal{U}_T} KL(D_S || D) \tag{1}$$

Before explaining the solution to the above optimization problem, we show the relationship between $D$ and $P(Y)$. To articulate our intuition, we use Figure 3(a) which illustrates a sample source distribution $D_S$ over 2D instance space (green). The labelling functions (in this case, $f_p$ and $f_a$) partitions the space of $\mathcal{X}$ into regions such that each region corresponds to a label combination $(y = \{y_p, y_a\})$. We denote each of these regions as $R_y$. Notice that $P(Y = y)$ becomes the integral of $D$ over the region $R_y$.

Let $\phi_S$ and $\phi_T$ be density functions of $D_S$ and $D_T$ respectively. Then, the above optimization problem becomes:

$$\phi_T = \arg\min_{\phi \in \Delta} \int \phi_S(x) \log \frac{\phi_S(x)}{\phi(x)} \, dx \qquad (2)$$

$$\text{s.t.} \int_{R_y} \phi(x) \, dx = P_T(y) \ \ \forall y$$

Where $\Delta \supset \mathcal{U}_T$ is the set of all distributions over $\mathcal{X}$ and region $R_y = \{x : f_p(x) = y_p, f_a(x) = y_a\}$. The Lagrangian equivalent for the above problem can be stated as,

$$L(\phi) = \int \phi_S(x) \log \frac{\phi_S(x)}{\phi(x)} \, dx + \sum_y \lambda_y \int_{R_y} \phi(x) \, dx,$$

where $\lambda_y \in \mathbb{R}$. Using the Euler-Lagrange equation to solve above problem,

$$\phi_T(x) = \phi_S(x) \times \frac{P_T(y)}{P_S(y)} \qquad (3)$$

Intuitively we find that $D_T^\star$ is a scaled version of $D_S$ with scaling factor as the ratio of $P_T(Y)$ and $P_S(Y)$. Note that this scaling factor may vary for different $x \in \mathcal{X}$ depending on which region $R_y$ it falls in. This is depicted in Figure. 3(b) which is a target distribution derived from source distribution in 3(a). The regions $R_{-1,+1}$ and $R_{+1,-1}$ are scaled up whereas regions $R_{+1,+1}$ and $R_{-1,-1}$ are scaled down. Though the above derivation is for two labels, one can see that Eq. 3 extends to any number of labels in $y$.

As mentioned earlier in this section, we have no instances from target domain. However $\phi_T(x)$ allows us to assign *soft domain labels* to the source domain instances, which indicates the probability of that instance belonging to the target domain. Specifically, let $y_D$ be the binary label indicating if an instance belongs to target domain. Then,

$$P(y_D|x) = \frac{\phi_T(x)}{\phi_T(x) + \phi_S(x)} = \frac{P_T(y)}{P_T(y) + P_S(y)}. \qquad (4)$$

Two possible assumptions that could be made on $P_T(Y)$ (if not provided) are uniform distribution over space of $y$ or uncorrelated (independent labels). Next we present two methods that leverage these soft domain labels.

### 3.3 ALGORITHMS

In the previous section we modeled negative transfer as a domain adaptation problem with soft domain labels. Next, we present methods to leverage the domain adversarial neural network (DANN) by Ganin & Lempitsky (2015) to induce domain-invariance features.

These methods are based on the theoretical analysis of domain adaptation by Ben-David et al. (2010; 2007). They provide a generalization bound on target error as following:

**Theorem 1** *(Ben-David et al. (2010)) Let $h \in \mathcal{H}$ be a prediction model of the form $h : \mathcal{X} \to \{-1, +1\}$ where $\mathcal{H}$ is the hypothesis space. Let $\epsilon_S$ and $\epsilon_T$ be generalization errors for source ($D_S$) and target ($D_T$) domains. For any classifier $h \in \mathcal{H}$*

$$\epsilon_T(h) \le \epsilon_S(h) + \frac{1}{2} d_{\mathcal{H}\Delta\mathcal{H}}(D_S, D_T) + \min_{h' \in \mathcal{H}} \epsilon_S(h') + \epsilon_T(h') \quad (5)$$

*where $d_{\mathcal{H}\Delta\mathcal{H}}$ denotes the $\mathcal{H}\Delta\mathcal{H}$-divergence between $D_S$ and $D_T$,*

$$d_{\mathcal{H}\Delta\mathcal{H}}(D_S, D_T) = 2 \sup_{h,h' \in \mathcal{H}} \left| \mathbb{E}_{x \in D_S}[h(x) \ne h'(x)] - \mathbb{E}_{x \in D_T}[h(x) \ne h'(x)] \right| \quad (6)$$

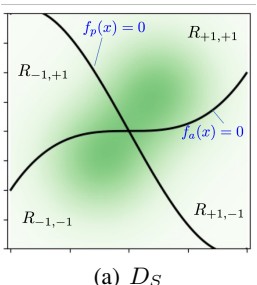

(a) $D_S$

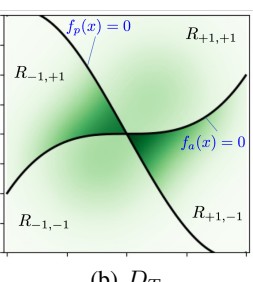

(b) $D_T$

Figure 2: An illustration of instance source distribution over 2D space.(a) Source distribution with correlated labels (b) Closest uncorrelated distribution is the desired target distribution.

Above theorem states that the target error is bounded by sum of source error and distance between source and target distributions. $d_{\mathcal{H}\Delta\mathcal{H}}$ can be seen as the maximum accuracy of classifying source and target domain by using a classifier from hypothesis space $\mathcal{H}\Delta\mathcal{H}$. Further, any classifier $g \in \mathcal{H}\Delta\mathcal{H}$ is the function XOR of some $h, h' \in \mathcal{H}$. DANN introduced an objective function which minimizes both source error as well as divergence between domains. Divergence between the domains can be looked at as the prediction accuracy of domain classifier.

DANN models starts with a mapping $J_f : \mathcal{X} \to \mathbb{R}^d$ with a parameter $\theta_f$, which projects instances to a latent representation space, we call this the *feature extractor*. These features are then mapped to primary label space with a mapping function $J_y$ (*label predictor*) with parameter $\theta_y$. The same features from latent representation space are mapped to domain label by $J_d$ (*domain classifier*), with parameter $\theta_d$. We denote the training set as $\{x^i, (y_p^i, y_D^i)\}_{i=1}^{N}$, with every instance we are provided with a primary label $y_p$ and a soft domain label $y_D$. The objective here is to find a feature extractor $J_f$ which projects instances to a latent representation space where *achievable* label prediction accuracy is high and domain prediction accuracy is low. Let $\mathcal{L}_y(\theta_f, \theta_y)$ be the prediction loss for label prediction and let $\mathcal{L}_d(\theta_f, \theta_d)$ be that for domain classifier, then objective function is

$$\theta_f^\star, \theta_y^\star, \theta_d^\star = \underset{\theta_f, \theta_y}{\arg\min} \; \underset{\theta_d}{\arg\max} \; \frac{1}{N} \sum_{i=1}^{N} \mathcal{L}_y^i(\theta_f, \theta_y) - \lambda \mathcal{L}_d^i(\theta_f, \theta_d), \tag{7}$$

where $\mathcal{L}_y(\theta_f, \theta_y)$ captures $\epsilon_S(h)$ and $\mathcal{L}_d(\theta_f, \theta_d)$ captures $d_{\mathcal{H}\Delta\mathcal{H}}(D_S, D_T)$ from Eq. 5. Note that $J_y \in \mathcal{H}$, then $J_d \in \mathcal{H}\Delta\mathcal{H}$. However, in using this formulation together with aforementioned soft domain labels, results in a weak $J_D$ since the soft domain labels are highly skewed towards the source domain. In such settings, $\mathcal{L}_d(\theta_f, \theta_d)$ on class imbalance domain labels no longer captures $d_{\mathcal{H}\Delta\mathcal{H}}(D_S, D_T)$. We address this issue again in Section 4.

The effort is to make $J_f$ provide a latent representation that is unable to discriminate source and target domain. As the soft domain labels are assigned according to $y = \{y_p, y_a\}$ (see Eq. 4), if that latent representation can be used to correctly predict $y_p$ and $y_a$ with label predictors $h_p \in \mathcal{H}$ and $h_a \in \mathcal{H}$, then there exists a $g \in \mathcal{H}\Delta\mathcal{H}$ which can predict source and target domains well (with $g(x) = h_p(x) \oplus h_a(x)$). Conversely, if the representations cannot be used to predict both the labels $y_p$ and $y_a$ correctly implies poor performance on domain classification. From this observation, we propose to replace domain classification loss (second loss term of Eq. 7) with auxiliary label classifier loss $\mathcal{L}_a(\theta_f, \theta_a)$, where $\theta_a$ is the parameter for the auxiliary label classifier $J_a$. We solve the optimization problem in Eq. 7 by using gradient reversal layer (Ganin & Lempitsky, 2015). Gradient reversal layer multiplies the gradient by a negative constant during the backpropagation. We call the auxiliary label classifier as an adversarial to the primary classifier. In this form, our idea is closely related to Zhang et al. (2018)

We extend this two label scenario to multilabels, by partitioning labels into groups, such that related labels are together. If a label is unrelated to all other labels than it forms a singleton group. We propose a model architecture, with a latent representation (with $J_f$) for each group. Further, the latent representation of a given group must be unable to predict any task from all other groups. We achieve this by adding all label classification other than the group member as the adversarial. In the next section, we showcase the empirical performance of multitask models trained in this way.

Another interesting alternative that departs from the usual view of feature projections is to utilize feature *selection* for $J_f$ (feature extractor). We discuss this alternative next.

### 3.3.1 FEATURE SELECTION

An intuitive method to prevent negative transfer between correlated features is to use a feature selection approach to explicitly select relevant features appropriate for a task.

We use Recursive Feature Elimination (RFE) (Guyon et al., 2002) for the task of feature selection using CNN as the classifier, since wrapper method (J. Tang, 2013) is computationally effective and noticeably time consuming when deep nets are used as the classifier. Recursive feature selection (Guyon et al., 2002) consider all the features and eliminated them at each iteration till the desired criterion is met. At each iteration, the current feature set is used to evaluate a task using a classifier. Each of the features obtain a score from the classifier, based on which one or more features are eliminated from the set. This step is repeated until the criterion is met, which can be in terms

of the final number of features to be chosen or desired classification performance to be attain. In order to perform RFE, we need to score each of the feature based on its effect on the classifier. For classifiers like logistic regression, the feature importance can be obtained by the weights assigned by the classifier for each of the features. In adversarial settings for multilabel classification, importance of the $k^{th}$ feature can be calculated as:

$$f^k = \frac{1}{|Y_p|} \sum_{i=1}^{|Y_p|} (w_{p_i}^k)^2 - \min\{(w_{a_1}^k)^2, \ldots, (w_{a_{|Y_a|}}^k)^2\} \tag{8}$$

where, $|Y_p|$ and $|Y_a|$ denote the number of labels to be predicted for the primary and the auxiliary task respectively.

## 4 EXPERIMENTAL ANALYSIS

### 4.1 SYNTHETIC EXPERIMENTS AND ANALYSIS

We study the adverse effects of negative transfer and how our proposed algorithms could resist this negative transfer, on synthetic data. Our synthetic dataset consists of 10-dimensional feature vectors, with two binary labels $y_p$ and $y_a$ (primary and auxiliary). We generate the data from a generative system with one class label distribution for training set and one for test set. First 5 features are sampled according to the primary label from a mixture of two Gaussian distributions with identity covariance matrix. If the primary label is 1 then 5 features are sampled from first Gaussian, otherwise from the second Gaussian. Similarly second set of 5 features are sampled same way from another set of mixture of two Gaussian distributions, this time sampled according to the auxiliary label.

Note that for these experiments we are only interested in predicting primary label $y_p$ in the test set. In all the experiments we keep $P(Y_p = 1) = P(Y_a = 1) = 0.5$. We also Keep distance between 2 Gaussian distributions corresponding to primary label to fixed to $1.5$. We use conditional label distribution $P(Y_p|Y_a)$ as measure label correlation. We compare following algorithms in our experiments: (1) **baseline** - A logistic regression classifier trained on primary label. (2) **DadvC** - One hidden layer (5 neurons) MLP having domain classification as the adversarial task. (3) **ALadvC** - Same as DadvC, with auxiliary label classification as the adversarial task. (4) **fs-adv** - Adversarial feature selection with auxiliary label classification as adversarial task.

Two sets of experiments where conducted to study the following aspects of negative transfer:

- Difference between train and test label correlation: For this experiment, we trained a model on training set with $P(Y_p|Y_a) = 0.8$ and tested out it's performance on test set which only differs from training set in $P(Y_p|Y_a)$. Figure 4.1(a) Shows the mean average precision of classifiers on test set. One can see that the baseline mAP drops as correlation in test set goes down. This indicates that baseline classifier captured wrong set of features for prediction. We can see that DadvC is performing marginally better than baseline. ALadvC and fs-adv are consistently performing well irrespective of varying label correlation in test. This shows that these methods picked the correct set of labels.
- Easiness of auxiliary task: We fix $P(Y_p|Y_a) = 0.8$ for training set and $P(Y_p|Y_a) = 0.5$ for test. We vary the distance between Gaussian distribution corresponding to auxiliary label while keeping that of primary label fixed. By doing this we vary the *easiness* of auxiliary task. In Figure 4.1(b) we can see that, as the auxiliary task gets easier, more features related to auxiliary task will be used by the primary classifiers which results in decreasing performance in test set. One can see that Both baseline and DadvC are performing bad as easiness increases. The proposed algorithms performs consistently better.

### 4.2 REAL DATA EXPERIMENTS AND ANALYSIS

**Datasets:** We use the Animals-with-Attributes (AwA) datasets (Xian et al., 2017) and Caltech-UCSD Birds 200-2011 (CUB) (Wah et al., 2011) for the multilabel attribute prediction task. With both datasets we follow the experimental protocol from (Xian et al., 2017). The Animals with Attributes-2 (AWA) consists of $30,475$ images of animals in natural settings, with 50 animals and 85 annotated attributes each. According to Jayaraman et al. (2014), 85 attributes can be grouped

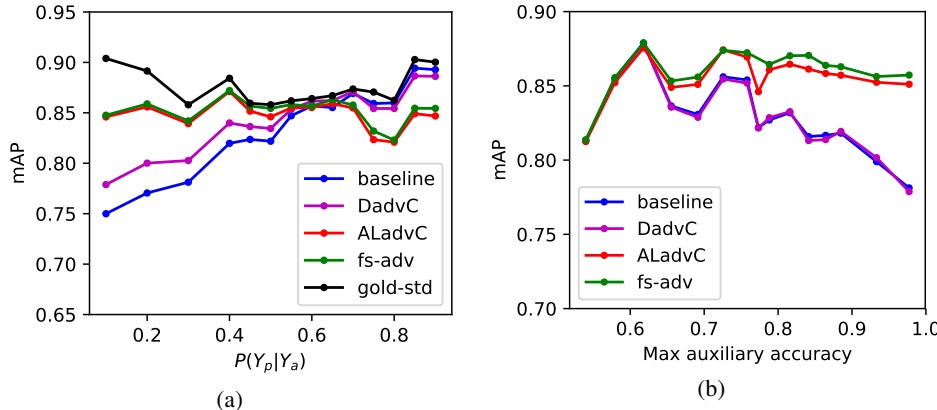

(a)            (b)

Figure 3: Synthetic experiment results:(a) Mean average precision over test set with varying $P(Y_p|Y_a)$, when trained on data with $P(Y_p|Y_a) = 0.8$. (b) Mean average precision over test set with $P(Y_p|Y_a) = 0.5$ when train set with $P(Y_p|Y_a) = 0.8$, varying max auxiliary accuracy.

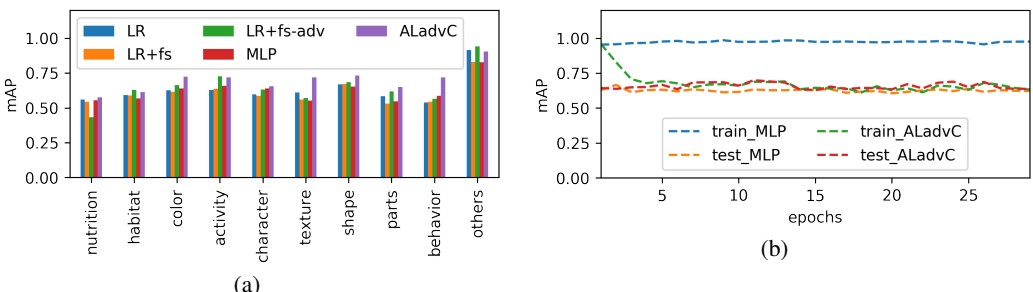

(a)            (b)

Figure 4: Mean average precision over test split of AwA per group. The proposed ALadvC out performs the MLP baseline model, while the LR+FS-adv outperforms its baseline LR and LR+FS, for multilabel attribute prediction on held-out classes. (b) The training performance of the model with adversarial loss is corrected to reflect the true performance of the model, improving generalization.

into 10 relevant groups. The dataset is split across the animals as 27/13/10 for train, validation, and test respectively as done in (Xian et al., 2017). The Caltech-UCSD Birds 200-2011 (CUB) consists of 11,788 images of birds, captured in natural settings, corresponding to 200 birds. The dataset also consists of 312 attributes of the birds annotated per image. They are aggregated into 28 groups corresponding to anatomical parts, and split across classes: 100/50/50 for train, validation and test over attributes, as done in (Xian et al., 2017). The split ensures different correlations between attribute labels in the three splits, which highlights the problems of negative-transfer.

**Feature representations**: We directly utilize the the feature representation (of length 2048) obtained from ResNet-101 (He et al., 2016) model that is pre-trained on ImageNet (Deng et al., 2009) for AwA and CUB. We explain below the specific architectures and parameters used for each experiment. Mean average precision (mAP) on validation and test sets are reported. Attribute prediction is an unbalanced prediction task, as individual attributes are rare. We use a balance corrected binary cross-entropy loss function for all experiments, with the balance count obtained from the training set. Further, we utilize early stopping criteria based on the performance of the model on the validation set.

**LR, LR+FS and LR+FS-adv**: As discussed in Section 3.3.1, one way to prevent the adverse effects of negative transfer is by selecting the optimal feature set for each of the task, based on the proposed adversarial objective function. We use Recursive Feature Elimination (RFE) (Guyon et al., 2002) method for the task of feature selection wrapped over a Logistic Regression (LR). We perform feature selection experimentation using Logistic Regression (instead of multi-layer perception, as

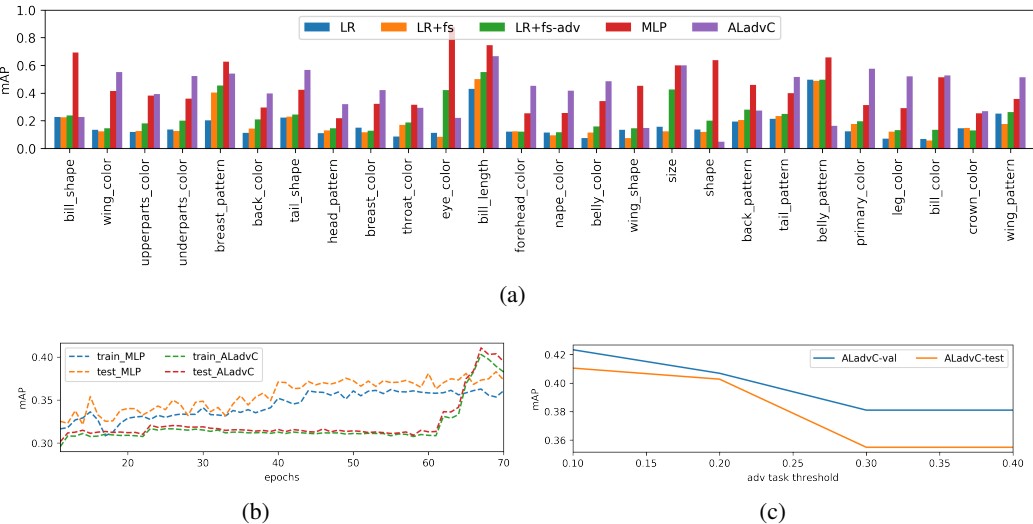

(a)

(b)                                                              (c)

Figure 5: Mean average precision over validation and test split of CUB per group. The proposed ALadvC out performs the MLP baseline model, while the LR+FS-adv outperforms its baseline LR and LR+FS, for multilabel attribute prediction on held-out classes. (b) mAP of MLP and proposed ALadvC model on train and test against epochs. The training performance of the model with adversarial loss is corrected to reflect the true performance of the model, improving generalization. (c) The CUB dataset consists of 28 groups, increasing number of adversarial tasks from most correlated to least helps improve performance on test splits.

Table 1: Table showing the improvement in mean average precision (%) using our proposed methods for AwA and CUB datasets.

| Approach | AwA | | CUB | |
|---|---|---|---|---|
| | val | test | val | test |
| LR | 65.27 | 63.22 | 16.00 | 15.98 |
| LR+FS | 66.41 | 61.13 | 24.80 | 17.57 |
| LR+FS-adv | 71.07 | 64.62 | 25.11 | 23.16 |
| MLP | 71.27 | 63.05 | 37.51 | 37.53 |
| ALadvC | 74.15 | 69.92 | 42.34 | 41.05 |

done in next set of experiments) for time efficiency. For CUB, we transform the 2048 features into a 500 dimensional space by using a dense layer followed by ReLU, which is then used for feature selection. Based on the primary task at hand, the feature importance scores are calculated using Eq. 8, by using the learned weights of LR. At each iteration, we remove $\lambda \times 0.33$ numbers of features, where we decrease $\lambda$ at each iteration as: $0.95 \times \lambda$. The final number of features to be retained is decided using the validation based on the mAP. We also observe the performance of feature selection without the adversarial setting (LR+fs). In this case, we only select the features that perform well for the primary task, without considering any effect on the auxiliary tasks. Performances of RFE have been observed in both adversarial and non-adversarial settings, which is reported in Table 1, and group-wise test mAP in Figures 4(a) and 5 (a) respectively.

As shown in Table 1, the mAP of RFE improves the baseline accuracy when LR has been used (compare rows 2 and 3 with row 1). The efficiency of the proposed criterion for scoring features, as shown by equation 8, can be observed when we compare between rows 2 and 3 in the Table. There is an improvement of 2.3% and 45% compared to LR (row 1), and 5.7% and 31.8% compared to LR+FS (w/o adv) (row 2) for AwA and CUB datasets respectively showcasing the advantage of the proposed method (LR+FS-adv). The average number of features selected are 321 (out of 2048) and 140 (out of 500) for AwA and CUB respectively. It is evident from the Table that the performance of MLP and ALadvC outperforms LR+FS with adv by a large margin. The performance can be attributed to MLP and ALadvS can apply projections to the feature vectors rather than selection/omission.

**MLP and ALadvC**: We test the performance of the proposed adversarial approach as described in Section 3.3. We utilize the representation vector obtained from ResNet-101 as the base model. Next, we attach a trainable layers (AwA:500, CUB:500‖100) with ReLU. Next, we add *latent* representation per group (AwA:10, CUB:5) with a linear connect to the group specific attribute prediction (with sigmoid activation). The smaller size of the latent layer ensures high feature transfer leading to more negative transfer. The baseline model (MLP) and the proposed model with auxiliary labels for each group as an adversarial classifier (ALadvC) are both trained with learning rate of $lr = 0.01$, which is decayed exponentially. Additionally, the gradient reversal weight $\lambda = \frac{K}{1+\exp{-10*l_i}}$, is related to the $i^{th}$ step of training, where $l_i = i/num\_steps$. The scheduler increases the weight exponentially towards K (described in (Ganin et al., 2016)). The model configuration (number and size of intermediate layers, $K$, $lr$) have all been picked from a large parameter sweep for best validation error. Due to the large number of adversarial branches (28 groups corresponding to 27 adversarial branches per group), we utilize the pairwise-task-label overlap to threshold the number of tasks (details below). Table 1 reports mean average precision for attribute prediction and Figure 4 (a) and 5 (a) show the group-wise performance of the proposed model compared to its corresponding MLP (best baseline model in parameter search).

ALadvC performance improves by 2.8% and a 6.8% mAP improvements on the validation and test splits for AwA, respectively. Similarly, 4.8% and 3.5% improvements are observed in CUB dataset. Note that the splits ensure different correlations between attributes in the validation and test splits. Further, 4 (b) and 5 (b), show the effect of the gradient reversal on the learning process as the $\lambda$ progressively improves. We notice a drop in the train mAP and simultaneous improvement over test mAP as the model

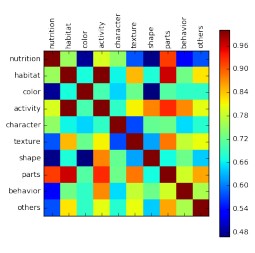

(a) AwA-Train

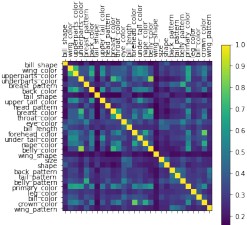

(b) CUB-Train

Figure 6: Maximum cross-correlation between the grouped attributes from train splits of AwA and CUB dataset.

sheds improvement obtained from negative transfer. Ultimately the performance on both datasets improves, suggesting the overall adverse effect of negative transfer. The group-wise performance measured on the test sets show drop in mAP for certain groups. We assert that the drop represents the true performance of the model on these tasks. As mentioned above, we identify an additional imbalance in the attribute prediction task, we term class-combination imbalance. The measure indicates the pairwise co-concurrence of attributes, which is computed as the pairwise Jaccard similarity between each attribute (illustration of the maximum Jaccard similarity cross group for both datasets in Figure 6, for their corresponding train splits). We utilize the measure to identify the subset of adversarial tasks per group for the CUB dataset. As shown in Figure 5 (c), the performance of ALadvC improves when all group tasks are utilized. The empirical result eludes to the advantage of the number or hardness of the adversarial task in improving prediction performance.

## 5    CONCLUSION AND FUTURE WORK

In this work, we show that adversarial learning is the natural answer to prevent negative transfer. This leads to potential improvement in any supervised learning of natural data that is seeking generalization. We find that even in relatively straight-forward linear models presented above, co-occurrence of unrelated labels hampers performance and must be explicitly treated. We address the problem of negative transfer in a multi-task scenario, and also show the applicability of our solution in any supervised task. Supervised learning practitioners can utilize domain expertise to acquire and leverage additional *negative* labels for this purpose. Recent work in explainability of machine learning models can also be appropriately leveraged to facilitate this task.

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
