# OpenReview forum: "REVISTING NEGATIVE TRANSFER USING ADVERSARIAL LEARNING"
_ICLR.cc/2019/Conference_

### Official Review · AnonReviewer1 · 2018-11-01
**Interesting application of adversarial learning to tackle negative transfer, but further analysis on source of performance improvement required**

**Rating:** 6
**Confidence:** 4

**Review:**

- The authors study the problem of negative transfer in representation learning, and propose to use the formulation proposed by Ganin & Lempitsky '15 for domain adaptation to reduce negative transfer. Instead of defining the domain classification as the adversarial task to learn a domain-independent representation, they collect a set of classification problems irrelevant to the main task as the adversarial tasks, and aim to learn a representation that focuses only on the primary task. There are very little changes compared to the proposal by Ganin & Lempitsky '15, but the application to solve the problem of negative transfer is interesting.

- My main concern on the whole argument of the paper is whether the benefits we see in the experiments come from the elimination of negative transfer, or just come from having more training labels from different tasks available. In the main formulation of the approach (equation 7), the authors try to learn a feature representation that works well for the primary task but works poorly for the auxiliary(irrelevant) tasks. If we switch the sign for lambda, then it becomes very similar to traditional multi-task learning. I wonder how the multi-task formulation would compare against the adversarial formulation proposed by the authors. There are reasons to suspect the multi-task formulation will also work better than the logistic regression baseline, since more labels from different tasks are available to learn a better joint representation. It is not clear whether the improvements come from modeling the auxiliary tasks using negative transfer (where the adversarial approach should beat the baseline and multi-task approach), or just come from having more information (where both the adversarial approach and the multi-task approach beat the baseline, but have similar performance).

- From a practical point of view, it is not easy to decide what prediction tasks are irrelevant. For example, in the birds dataset, I would expect the color and patterns in the body parts to have some correlations (primary_color, upperparts_color, underparts_color, wing_color, etc). In the case of occlusion of the relevant body parts, I could make a guess on the color based on the colors on other parts of the bird. In the ideal case for the current method, I would expect the adversarial approach proposed to learn a representation that mask out all the irrelevant parts of the animal or irrelevant contextual information. Apart from showing improved prediction performance, have the authors perform analysis on the image activation patterns similar to the motivation example in Figure 1 to see if the new approach actually focus on the relevant body parts of the animals?

- The definition of auxiliary tasks are described in the second last paragraph of 3.3, but it would be clearer if it is also mentioned how they are defined in the experiments section. I went through the whole experiments section having trouble interpreting the results because I could not find the definition of adversarial tasks.

- Overall I like this paper since it attempts to solve an interesting problem in computer vision, but I would like to see the above question on comparison with multi-task learning answered, or some image activation pattern analysis to provide a more solid argument that the improvements come from elimination of negative transfer.

---

### Official Review · AnonReviewer3 · 2018-11-02
**Interesting formulation; lack of mention and comparison to related work, terminology issue, and other flaws**

**Rating:** 2
**Confidence:** 4

**Review:**

Pros:
- Provides illustration and math formulation for the problem of generalization beyond the correlation of labels and correlated but irrelevant attributes. Forming the issue as a domain adaptation problem (or specifically, a special kind of probability shift) is a clever idea.


Cons:
- Lacks comparison to existing work. Making features invariant to attributes to improve generalization is not a new idea, cf. :
(1) Xie, Qizhe, et al. "Controllable invariance through adversarial feature learning." Advances in Neural Information Processing Systems. 2017.
(2) If you consider the domain difference between various domains to be similar to attribute, then this is also related: Li, Haoliang, et al. "Domain generalization with adversarial feature learning." Proc. IEEE Conf. Comput. Vis. Pattern Recognit.(CVPR). 2018.
(3) There are other works that, although do not aim at improving generalization, use very similar formulation to decouple attribute from features: e.g. (a) Lample, Guillaume, et al. "Fader networks: Manipulating images by sliding attributes." Advances in Neural Information Processing Systems. 2017.  (b) Mitigating Unwanted Biases with Adversarial Learning (which the authors cite, but do not offer any comparison or differentiation)
To improve the paper, these related work should be discussed in related work section, and (if applicable) compared to the proposed method in the experiments, rather than a very brief mention of one of them in Section 3.3 and no comparison.

- Use of the term "negative transfer" is problematic. This is a more important shortcoming, but people may disagree with me. As far as I know, this term is used to describe a *source task* being used to help a *different target task* but result in a negative gain in performance (Torrey, Lisa, and Jude Shavlik. "Transfer learning."), which is inherently a multi-task learning setting. However, in this paper it refers to the effect of unrelated features being used in classifier, resulting in a worse generalization. The existence of this issue does not involve a second task at all. If this is not intended, please use another phrase. If the authors think that these are one and the same, I would strongly argue against this proposition.
Also, there is no "negative transfer technique" as implied by page 2, end of the first paragraph.

- Section 3.2 and 3.3's analysis is somewhat disjoint from the method. The analysis boils down to "given a different correlation between primary and aux tasks, you can compute the distribution of inputs, which will be different from the source, so let's make the aux task unpredictable to get domain invariance." And the method goes on to remove auxiliary task information from the shared feature space. This is disjoint from either eq. (1) picking a target domain closest to source, and Theorem 1 the bound for domain adaptation. One way to improve the paper is to analyze how these analysis are affected by the adversarial training.

- One of the selling points is that the method can adapt to trainable features in deep learning. However, in the experiment, fixed extracted features from pre-trained ResNet is used anyway. If so, a way to improve the paper is to compare to the traditional methods cited in page 2 paragraph 1, by applying them on fixed extracted ResNet features.

---

### Official Review · AnonReviewer2 · 2018-11-02
**The problem setting is strange, and the assumptions used in the proposed algorithms are too strong**

**Rating:** 4
**Confidence:** 4

**Review:**

The term "negative transfer" is quite confusing, especially when it is used together with the term "domain adaptation". In domain adaptation, negative transfer means transferring knowledge from a source domain to a target domain in a brute-force manner may result in worse performance compared with that obtained by only using the target domain data.
In this paper, the negative transfer problem is different from that in domain adaptation. The authors just tried to model the proposed negative transfer learning problem as a domain adaptation problem. However, the defined problem setting of negative transfer is quite strange, where for the target dataset, neither instances nor labels are available expect for the probability of P_T(Y_p, Y_a), and there is relationship between Y_p and Y_a, which is different from that of the source dataset. It is not convincing that why the proposed problem setting is important in practice.

The proposed algorithm is designed based on two strong assumptions:
1. D_T is drawn from a distribution that is nearest to that of D_S, and
2. P_T(Y) is given in advance.
Regarding the first assumption, it is not reasonable, and it is hard to be satisfied in practice. For the second assumption, it is also too strong to be satisfied in practice. Though the authors mentioned that when P_T(Y) is not given in advance, P_T(Y) can be further assumed to be of the uniform distribution or the classes are uncorrelated. However, these are just ad-hoc solutions. In practice, if P_T(Y) is unknown, and it is very different from the uniform distribution, or labels are highly correlated, the proposed algorithm may perform very poorly.

Regarding the details of the algorithm, it just simply applies an existing model, DANN. In addition, the theoretical part is a well-known theorem.

There are some typos: on Page 3, Figure 3(a) --> Figure 2(a); on Page 4, Figure 3(b) --> Figure 2(b).

---

### Author Response · Authors · 2018-11-27
**Thanks for your comments**

We thank the reviewers for the constructive remarks on our idea and pointing out some relevant literature. Based on your comments, we will revamp the presentation and articulation of the paper for a future venue.

The concerns of the reviewers can be summarized as follows with our brief response (to foster a discussion):

-- Why we chose to use the term "negative transfer"?
We used the term "negative transfer" to describe the problem emerging from correlated tasks in multi-task learning which has been looked at in earlier papers, especially in [1]. These papers propose using regularization [1,5] to prevent sharing features among tasks which are not related to each other. Further, we find this problem is not confined to multi-task learning but can extend to any supervised learning approach. This problem setting was previously addressed as "negative transfer" in [2,3,4].

It seems "negative transfer" term has different meaning in domain adaptation. We agree with the reviewers that there is an ambiguity in the meaning of "negative transfer" in the community. We shall explicitly address this in a future submission.

-- Differentiate from DANN (Ganin & Lempitsky '15)
DANN is a domain adaptation technique that uses gradient reversal to explicitly prevent encoding of domain information in the feature representation. As discussed in Section 3.2 (paragraph 2), we do not have access to unlabeled instances of a "target domain" in this work. Further, we deal with large number of tasks and corresponding adversarial tasks (here AwA dataset: 85 attributes, CUB dataset: 312 attributes) by using a novel adversarial task weighting scheme together with gradient reversal proposed by Ganin and Lempitsky'15.

To reiterate the contributions of this work,
 - We draw the attention of the community to "negative transfer" problem that was previously looked at before the deep learning era in all supervised learning problems. Further, we show the limitation of previously proposed approaches to tackle negative transfer using various regularization methods.
 - We pose the negative transfer problem as an instance of domain adaptation with strong assumptions. We show that DANN can then be used to prevent negative transfer in this setting. We then propose a feature selection variant of adversarial learning, that can also tackles negative sharing.
 - We empirically show improvement in attribute prediction (on unknown classes)  on two public datasets over a known protocols [1].

-- Clarification on the experimental protocol
 - our models use ResNet101 as a base model with one trainable layer (of size 500).
 - baseline model is multitask learning without adversarial arms.
 - attribute prediction using correlation does not generalize (body-color and wing-color correlation may not be applicable to a different bird)
 - we shall have illustrations with visualization of activation patterns in a future draft.

References:
[1] Dinesh  Jayaraman,  Fei  Sha,  and  Kristen  Grauman. Decorrelating  semantic  visual  attributes  by resisting  the  urge  to  share.  CVPR 2014
[2] Lee, Giwoong, Eunho Yang, and Sung Hwang. Asymmetric multi-task learning based on task relatedness and loss. International Conference on Machine Learning. 2016.
[3] Hae Beom Lee, Eunho Yang, Sung Ju Hwang. Deep Asymmetric Multi-task Feature Learning. International Conference on Machine Learning. 2018.
[4]  Ruder, Sebastian. An overview of multi-task learning in deep neural networks. arXiv preprint arXiv:1706.05098 (2017).
[5] Yang Zhou, Rong Jin, and Steven Chu-Hong Hoi.  Exclusive lasso for multi-task feature selection. In Proceedings of the Thirteenth International Conference on Artificial Intelligence and Statistics, (AISTATS) 2010

---

### Meta-Review · Area_Chair1 · 2018-12-11
**Refinement of objective and comparison against prior work needed**

**Confidence:** 4
**Recommendation:** Reject

**Metareview:**

This paper proposes reducing so called "negative transfer" through adversarial feature learning. The application of DANN for this task is new. However, the problem setting and particular assumptions are not sufficiently justified. As commented by the reviewers and acknowledged by the authors there is miscommunication about the basic premise of negative transfer and the main assumptions about the target distribution and it's label distribution need further justification. The authors are advised to restructure their manuscript so as to clarify the main contribution, assumptions, and motivation for their problem statement.

In addition, the paper in it's current form is lacking sufficient experimental evidence to conclude that the proposed approach is preferable compared to prior work (such as Li 2018 and Zhang 2018) and lacks the proper ablation to conclude that the elimination of negative transfer is the main source of improvements.

We encourage the authors to improve these aspects of the work and resubmit to a future venue.